# The Relationship between Exertional Desaturation and Pulmonary Function, Exercise Capacity, or Medical Costs in Chronic Obstructive Pulmonary Disease Patients

**DOI:** 10.3390/medicina59020391

**Published:** 2023-02-17

**Authors:** Meng-Lin Tsai, Chin-Ling Li, Hui-Chuan Chang, Yuh-Chyn Tsai, Ching-Wan Tseng, Shih-Feng Liu

**Affiliations:** 1Division of Pulmonology and Critical Care Medicine, Department of Internal Medicine, Kaohsiung Chang Gung Memorial Hospital, Kaohsiung City 833, Taiwan; 2Department of Respiratory Therapy, Kaohsiung Chang Gung Memorial Hospital, Kaohsiung City 833, Taiwan; 3College of Medicine, Chang Gung University, Taoyuan 333, Taiwan

**Keywords:** COPD, exertional desaturation, ADO index, BODE index, medical costs

## Abstract

*Background and Objectives:* Exertional desaturation (ED) is common and is associated with poorer clinical outcomes in chronic obstructive pulmonary disease (COPD). The age, dyspnea, airflow obstruction (ADO) and body mass index, airflow obstruction, dyspnea, and exercise (BODE) indexes are used to predict the prognosis of COPD patients. This study aimed to investigate the relationship between these indexes, pulmonary function, medical costs, and ED in COPD patients. *Materials and Methods:* Data were collected from the electronic database of the Kaohsiung Chang Gung Memorial Hospital. This retrospective study included 396 patients categorized as either ED *(n* = 231) or non-ED (*n* = 165). Variables (including age, smoking history, body mass index (BMI), pulmonary function test, maximum inspiratory pressure (MIP) and maximum expiratory pressure (MEP), six minutes walking test distance (6MWD), SpO_2_, COPD Assessment Test (CAT) score, ADO index, BODE index, Charlson comorbidity index (CCI), and medical costs) were compared between the two groups, and their correlations were assessed. ED was defined as SpO_2_ less than 90% or SpO_2_ decrease of more than 4% compared to baseline levels during 6MWT. *Results:* A significant statistical difference was found regarding a lower score of the ADO index and the BODE index (both *p* < 0.001), better pulmonary function (forced expiratory volume in the first second (FEV1), *p* < 0.001; FEV1/ forced vital capacity (FVC), *p* < 0.001; diffusion capacity of the lung for carbon monoxide (DLCO), *p* < 0.001), and higher minimal oxygen saturation (*p* < 0.001) in non-ED COPD patients. No difference was found in the distance of the 6MWT (*p* = 0.825) and respiratory muscle strength (MIP; MEP, *p* = 0.86; 0.751). However, the adjusted multivariate logistic regression analysis showed that only SpO_2_ (minimal) had a significant difference between of the ED and non-ED group (*p* < 0.001). There was either no difference in the medical expenses between ED and non-ED COPD patients. *Conclusions:* SpO_2_ (minimal) during the 6MWT is the independent factor for ED. ED is related to BODE and ADO indices, but is not related to medical expense.

## 1. Introduction

Chronic obstructive pulmonary disease (COPD) results from chronic inflammation with narrowing of the small airways, reduction in the number of small airways (including alveolar ducts, alveolar sacs, and alveoli), an increased number of goblet cells, and mucus gland hyperplasia. Thus, chronic bronchitis and pulmonary emphysema are the two main presentations of COPD patients [1]. In addition, these patients often have a clinical history of dyspnea, chronic cough with sputum, chest tightness, decreased physical activity, and desaturation. Therefore, a diagnosis of COPD requires knowledge of the related risks, presenting signs and symptoms, and a pulmonary function test (PFT) [1,2,3].

The PFT is a widely used tool to measure parameters of lung function, such as forced expiratory volume in the first second (FEV1), forced vital capacity (FVC), and diffusion capacity of the lung for carbon monoxide (DLCO). PFT differentiates between obstructive and restrictive respiratory disorders; PFT is also used to categorize the severity of airflow limitation in COPD patients (from GOLD I to GOLD IV) [2,3]. The 6-min walking test (6MWT) and its walking distance (6MWD) are often used to measure COPD patients’ functional exercise capacity. The 6MWT also measures coexisting extrapulmonary manifestations of chronic respiratory disease, including cardiovascular disease, fragility/weakness, and sarcopenia [4].

Both the ADO (age, dyspnea scale, obstructive airflow) index (or serially ADO score change) and the BODE (body mass index, obstructive airflow, dyspnea scale, exercise capacity) index can be used to predict the 3-year or longer mortality rate and prognosis in COPD patients. Some studies also used e-BODE (BODE plus exacerbations) to predict the 6-month mortality rate in these patients [5,6,7,8]. Moreover, the Charlson comorbidity index (CCI or CC index) is a criterion for 19 chronic diseases. A high CCI indicates a higher number of chronic diseases, increased severity of comorbidity, higher treatment costs, and a higher mortality rate [9,10]. COPD patients often have coexisting comorbidities or chronic systemic diseases that further deteriorates their activity capacity or increases their medical costs (hospitalization).

Exertional desaturation (ED) is common and is associated with poorer clinical outcomes in COPD [11,12,13]. ED is also a predictor of rapid decline in lung function in patients with COPD [14,15]. Many studies have assessed the severity and prognosis of COPD patients, using these tools (ADO index, BODE index, e-BODE, or CCI) to evaluate the outcome of COPD patients. In addition, desaturation is another presentation in COPD patients during the 6MWT; this combination was referred to as ED in this study. However, the role of ED in COPD patients and its relationship with the ADO index, BODE index, CCI, COPD-related symptoms, pulmonary function, exercise capacity, and medical costs (hospitalization) remains unclear. Thus, this study aimed to examine the relationship between these factors in COPD patients.

## 2. Materials and Methods

### 2.1. Study Design

In this retrospective study, the clinical outcomes of ED (defined as SpO_2_ less than 90% or SpO_2_ decrease by more than 4% compared to baseline levels during the 6MWT [15,16]) in COPD patients were investigated. The enrolled patients with COPD into two groups: the ED group and the non-ED group. Variables (including age, smoking history, BMI, PFT, MIP and MEP, 6MWD, SpO_2_, CAT score, ADO index, BODE index, CCI, and medical costs) were compared between the two groups, and their correlations were assessed.

### 2.2. Study Data

Data were collected from the electronic database of the Kaohsiung Chang Gung Memorial Hospital (KCGMH). Patients who underwent a 6MWT in the respiratory therapy department (*n* = 1063) between 2015 and 2016 were selected for data collection and analysis. The 6MWT reports of these patients were dated from 31 January 2015 to 31 August 2017 (32 months). The electronic medical records and healthcare resource utilization were sourced from KCGMH.

The Inclusion criteria of patients with COPD in KCGMH were as following: (1) diagnostic code ICD-9-CM:490~496 or ICD-10-CM:J41~J44, meeting the diagnosis of chronic bronchitis or pulmonary emphysema [1] or COPD (*n* = 507); (2) with a complete record of 6MWT distance (6MWD); (3) diagnosis of COPD by a post-bronchodilator test result (FEV1/FVC < 70%) with persistent expiratory airflow obstruction according to the COPD GOLD guideline [2]; and (4) patients older than 40 years old. The medical information of all selected patients (*n* = 396) was collected. The 396 selected patients were then divided into two groups (ED group (*n* = 231) and non-ED group (*n* = 165)) based on whether or not the patients had ED during the 6MWT. The study process is shown in Figure 1.

The BODE index was categorized into four quartiles: quartile one used a score of 0~2; quartile two used a score of 3~4; quartile three used a score of 5~6; and quartile four used a score of 7~10 [10,11]. CCI data were obtained from the patient’s retrospective reports diagnostic codes during the 32 months. The data on healthcare resource utilization consisted of total medical expenses, including outpatient clinic visits, hospitalization days, examination fees, and drug costs during the 32 months of each patient’s retrospective report. In addition, the correlations of the previously mentioned variables between the ED group and the non-ED group were analyzed.

### 2.3. Performance of 6MWT

The 6MWT is conducted indoors, along a long, flat, enclosed rectangular corridor with a 6 × 12 m area, a hard surface, and little traffic. The length of the corridor is marked every 3 m. Brightly colored tape at the turn-back points is marked on the floor. The required equipment include a countdown timer, mechanical lap counter, worksheets on a clipboard, a source of oxygen, oxygen nasal cannula, modified Borg Scale, pulse oximeter, and sphygmomanometer. We made our patients walking on the enclosed area in circles for total six minutes by themselves according to the instruction, then recorded their oxygen saturation, heart rate, and modified Borg Scale. Oxygen was prepared in case of our patients presenting respiratory distress during testing.

### 2.4. Ethical Approval

Patients were not required to give informed content to review their medical records as this was a retrospective study. Even so, the study complied with the Helsinki Declaration and Good Clinical Practice Guidelines and was approved by the Ethics Committee of KCGMH (IRB: 201701293B0).

### 2.5. Statistical Analysis

Data distributions are presented as the mean and standard deviation (mean ± SD), median (interquartile range, IQR), or N (%). First, Chi-square tests, Fisher’s exact test, and one-way analysis of variance (one-way ANOVA) were applied to compare the differences in variables by quartiles. Thereafter, posterior comparisons were made with Scheffé’s test. Lastly, one-way ANOVA and linear contrasts were applied to check for linear trends. Adjusted multivariate logistic regression analysis of the ED and non-ED group was also performed. Data analysis was performed using the Statistical Package for the IBM Social Sciences (SPSS) version 26 software (New York, NY, USA).

## 3. Results

A total of 396 COPD patients (382 (96.5%) male, average age 71.3 ± 8.4 years) met the inclusion criteria and were included in this study. Table 1 shows the baseline characteristics of the studied population. The average duration of smoking was 31.7 ± 18.5 pack-years, while the average body mass index (BMI, kg/m^2^) was 23.5 ± 4.1.

Regarding the pulmonary function test (PFT), their FEV1/FVC showed obstructive type. The diffusion capacity of the lung for carbon monoxide (DLCO) was implying the presence of pulmonary emphysema in our patients. The mMRC dyspnea scale, ADO index, BODE index, CCI, GOLD stage, and 6MWD are also presented in Table 1

After analyzing the ED and non-ED groups, the results of which are shown in Table 2, we found no difference in patient age, smoking history, and BMI between the two groups (*p*-value: 0.465; 0.751; 0.127, respectively). On the other hand, we found a significant difference in FEV1 (% of predicted value), FEV1/FVC, and DLCO between the two groups (*p*-value < 0.001). There was no difference between the respiratory muscle strength of patients, or the MIP and MEP, between the two groups (*p*-value: 0.86; 0.751, respectively). The mMRC scale was lower in the non-ED group than in the ED group (*p*-value: 0.028). Similar results were seen with the CAT score. Although there was no difference in the 6MWD and the SpO_2_ (initial) between the two groups (*p*-value: 0.825; 0.129, respectively), there was a significant decrease in SpO_2_ (minimal) in the ED group compared to the non-ED group (*p*-value < 0.001). However, adjusted multivariate logistic regression analysis showed that only SpO_2_ (minimal) had significant difference between of the ED and non-ED group (*p*-value < 0.001).

In reference to medical expenses, listed in Table 3, no difference was noted in the number of outpatient visits, outpatient medical expenses, number of hospitalization days, hospitalization medical expenses, and total medical expenses between the ED and the non-ED groups (*p*-value: 0.92; 0.45; 0.13; 0.36; 0.41, respectively). The ADO index (*p*-value < 0.001) and BODE index (*p*-value < 0.001) scores were significantly lower in the non-ED group compared to the ED group. BODE quartiles 3 and 4 and ADO quartiles 3 and 4 were related to ED group (*p*-value < 0.001). GOLD stage was also related with ED (*p*-value < 0.001). Even so, no difference was noted in CCI between the two groups (*p*-value: 0.65); the results are shown in Table 3. Figure 2 presents the ADO and the BODE index distribution in both the ED group and non-ED group and their correlations.

## 4. Discussion

Our study demonstrated that SpO_2_ (minimal) during the 6MWT is the independent factor associated with ED in COPD. Resting oxygen saturation, lung function, and DLCO cannot reliably predict which patients with COPD will experience ED. ADO and BODE indices are related to ED. ADO and BODE indices are multiple factors combination predictor for COPD. ED may not be predicted by a single factor (FEV1 or DLCO). Because GOLD stage was stratified by FEV1, FEV1 showed no significance between ED and non-ED group by multivariate analysis. Therefore, we thought that GOLD stage was not reliable to predict ED.

The causes of exercise-induced desaturation in COPD patients are multifactorial, such as ventilation-perfusion mismatch, diffusion-type limitation, shunts, and decreased mixed venous oxygen content all contributing to a certain extent. Neither resting oxygen saturation nor pulmonary function studies can reliably predict which patients with COPD will develop ED [17], which is consistent out results. However, DLCO is important factor for ED in their study [17], which is not compatible with our study.

The degree of emphysema and co-existing comorbidities, such as heart failure and pulmonary hypertension, may be associated with ED. However, the study is retrospective, and most patients had no chest computed tomography, echocardiography, and right heart catheterization data. However, we suggest that COPD patients with ED should be evaluated by these examinations.

The severity of airflow limitation in COPD patients can be divided into four grades (GOLD I to GOLD IV) by the level of FEV1 (% of predicted value). Dyspnea severity of COPD (mMRC) and acute exacerbation (AE) times are also used as standards of measurement in these patients [2]. FEV1 is not an independent factor to cause ED by our statistics, and it maybe that COPD is a complex disease composed of multidimensional disorders [8] and is complicated by many co-existing comorbidities. The majority of the patients in this study were in GOLD II and small minorities were in GOLD IV, but there are more patients with ED than those without ED. It may be that most of these patients have other underlying diseases, such as heart failure, pulmonary hypertension, etc. Lower SpO_2_ in ED group may be the comprehensive presentation that is caused by all co-existing comorbidities. This study showed that a lower minimal SpO_2_ is strongly related to ED, which was consistent with Liu et al.’s study [13]. The ED group of COPD patients also have a positive relationship with the number of hospitalization days, although non-significant, which may be due to these patients being hospitalized because of other comorbidities, rather than simply COPD alone [18].

There was no difference in CCI observed between the ED and non-ED groups in this study. The fact that the score of CCI is the sum of 19 items (including the nervous system, cardiovascular system, respiratory system, digestive system, urinary system, endocrine system, hematology system, immune system, and cancer, with scores from 1 to 6) may be responsible for the lack of differences observed [9,10]. Additionally, the presence of comorbidities or systemic diseases other than respiratory system diseases in these patients might have contributed to the lack of observed difference. Common presenting conditions, such as decompensated heart failure, severe infection, gastrointestinal disorder, worsening kidney function, and uncontrolled cancer, were responsible for hospitalizations or deaths in our patients [9,10,19]. On the other hand, most of our patients received long-term care in our outpatient department. Some patients sought care from local clinics for medication or admission for COPD-AE treatment; our study did not include these medical expenses. Nonetheless, higher medical expenses would be expected in the ED group of COPD patients if their actual medical expenses at the various medical facilities they visited were included in our calculations.

Many COPD patients have complications such as chronic heart failure or pulmonary hypertension, which may cause desaturation, worsening functional capacity, frequent hospitalizations, and poorer outcomes [20,21,22,23]. ED is often observed in severe heart failure or pulmonary hypertension patients with poor outcomes [20,21,22,23]. Patients with severe heart failure or pulmonary hypertension are also expected to have higher ADO and BODE indexes. Even so, we found that even with a mean BMI larger than 21, there was no difference in the MIP/MEP between groups. Thus, according to their mean BMI, we assessed these patients as having sufficient muscle mass. Muscle strength (MIP and MEP) was, therefore, not a crucial factor for ED.

There were some limitations in our study. First, this was a retrospective study. Most patients did not have chest computed tomography, echocardiography, and cardiac catheterization data. Second, the studied data were collected from a single medical center, reducing the study’s objectiveness; thus, the study findings may not be representative of all COPD patients. Third, the majority of the patients were male. Sex differences may affect outcomes, disease management, and social health costs. Most of our enrolled COPD patients during that period were male, and we did not specifically exclude females. More than 80% of the COPD patients in Taiwan are male [24,25], which can be attributed to the fact that the number of women who smoke in Taiwan is far less than in various nations and states in the regions of Europe, North America, and Oceania. Fourth, the standard diagnosis of pulmonary hypertension requires a cardiac catheter to measure the wedge pressure of the pulmonary artery. This is an invasive procedure, so that few patients received this procedure. Echocardiography was performed in some of our COPD patients, but not all of them underwent the examination; therefore, we did not collect this data in our study. Fifth, medical costs can vary based on each doctor’s treatment. In 1995, Taiwan launched the single-payer National Health Insurance (NHI) plan. As of 2014, more than 99% of Taiwan’s population was admitted. The National Health Insurance Bureau of Taiwan has used the Diagnosis-Related Group (Tw-DRG) since January 2010. There is a uniform payment standard for medical expenses. The hospital subsequently claims the relevant fees from the government [26]. According to the DRG disease review system, the Medical Insurance Bureau pays the hospital a reasonable figure for its medical expenses (not necessarily 100%). In addition, our hospital is a large medical center in Taiwan and implements the attending physician system. The attending physicians are all professionally trained pulmonologists. The COPD patients in this study were all patients of the attending pulmonologist. Thus, each patient should theoretically receive the same treatment from the attending physician. Fifth, neither chronic heart failure nor pulmonary hypertension was explored in our patients. Therefore, any significant effects due to desaturation, worsening physical activity, increased hospitalizations, and poorer outcomes in our patients from these comorbidities were unaccounted for in this study.

## 5. Conclusions

SpO_2_ (minimal) during the 6MWT is an independent factor for ED. Resting oxygen saturation, lung function, and DLCO cannot reliably predict ED. ADO and BODE indices are related to ED. The ED group had higher medical expenses than the non-ED group, but it was not significant.

## Figures and Tables

**Figure 1 medicina-59-00391-f001:**
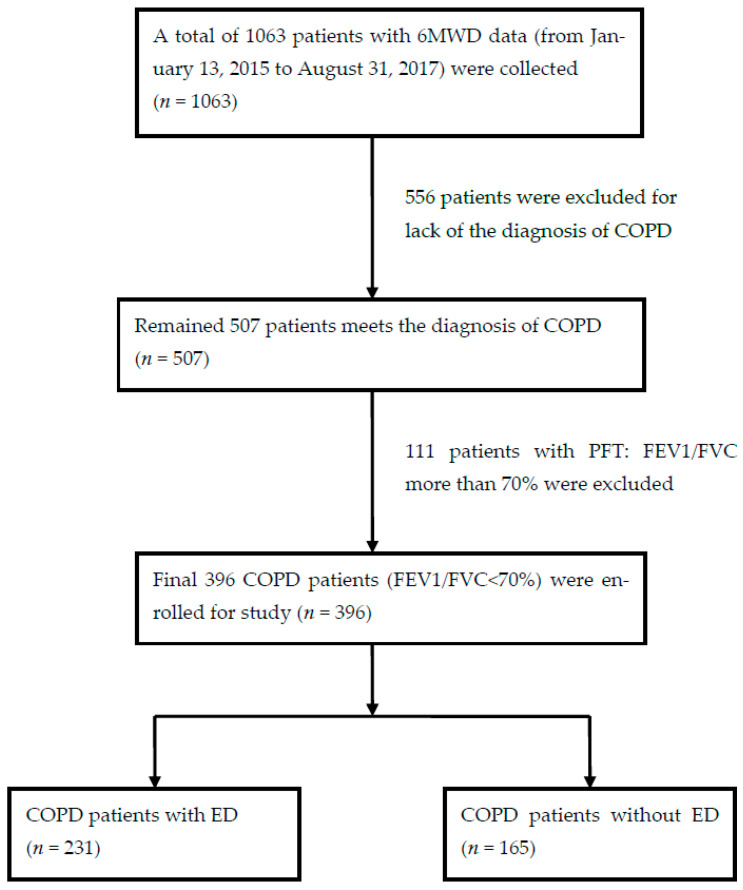
Flow chart of studied patient selection. Abbreviations: 6MWD, six minutes walking test distance; COPD, chronic obstructive pulmonary disease; FEV1, forced expiratory volume in 1st second; FVC, forced vital capacity; ED, Exertional desaturation.

**Figure 2 medicina-59-00391-f002:**
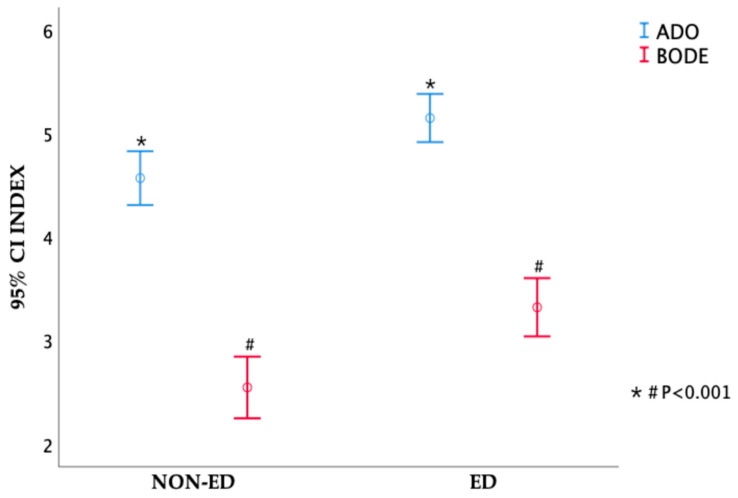
The difference of ADO index and BODE index between ED group and non-ED group.

**Table 1 medicina-59-00391-t001:** Baseline Characteristics of Enrolled 396 COPD Patients.

Factors	Mean ± SD or N (%)
Male (%)	382 (96.5)
Age (year)	73.1 ± 9.5
Smoking history (pack-year)	31.7 ± 18.5
Body mass index (BMI)	23.5 ± 4.1
FVC (% of predicted value)	79.7 ± 16.7
FEV1 (% of predicted value)	55.2 ± 18.2
FEV1/FVC (%)	52.7 ± 10.6
DLCO (%)	68.5 ± 21.0
mMRC	1.72 ± 0.9
mMRC dyspnea scale	
Scale 0 (%)	25 (6.3)
Scale 1 (%)	133 (33.6)
Scale 2 (%)	173 (43.7)
Scale 3 (%)	56 (14.1)
Scale 4 (%)	9 (2.3)
Charlson comorbidity index (CCI)	3.3 ± 2.8
BODE index	3.0 ± 2.1
ADO index	4.9 ± 1.8
GOLD stage (%)	
GOLD I (%)	46 (11.6)
GOLD II (%)	187 (47.2)
GOLD III (%)	140 (35.4)
GOLD IV (%)	23 (5.8)
MIP (cm-H_2_O)	72.2 ± 30.5
MEP (cm-H_2_O)	98.3 ± 46.8
6MWD (m)	351.9 ± 111.6

Abbreviations: FEV1, forced expiratory volume in 1st second; FVC, forced vital. capacity; DLCO, diffusion capacity of lung for carbon monoxide; MRC score, medical research council dyspnea scale; BODE index, composite index of body mass index, obstruction of airflow, dyspnea, and exercise capacity; ADO index, composite index of age, dyspnea, and obstruction of airflow; GOLD, Global Initiative for Chronic Obstructive Lung Disease; MIP, maximum inspiratory pressure; MEP, maximum expiratory pressure; 6MWD, 6 min walking test distance.

**Table 2 medicina-59-00391-t002:** Lung Function and Exercise Capacity of the ED and Non-ED Group.

Factors	ED Group (*n* = 231)	Non-ED Group (*n* = 165)	*p*-Value
Age (year)	73.0 ± 9.1	73.6 ± 8.7	0.465
Smoking history (pack-year)	31.9 ± 19	31.3 ± 18.1	0.751
Body mass index (BMI)	23.3 ± 4.1	23.9 ± 3.9	0.127
FVC (% of predicted value)	78.3 ± 18.0	81.6 ± 14.4	0.057
FEV1 (% of predicted value)	52.2 ± 19.3	59.5 ± 15.8	<0.001
FEV1/FVC (%)	50.7 ± 10.8	55.6 ± 9.6	<0.001
DLCO (%)	64 ± 21.5	75 ± 18.6	<0.001
mMRC score	1.8 ± 0.8	1.6 ± 0.9	0.028
MIP (cm-H_2_O)	72.4 ± 31.6	71.9 ± 29	0.86
MEP (cm-H_2_O)	97.6 ± 37.9	95.6 ± 30.2	0.751
6MWD (m)	350.9 ± 113.9	353.5 ± 108.6	0.825
SpO_2_ (initial)	96.1 ± 3.1	96.5 ± 1.7	0.129
SpO_2_ (minimal)	88.6 ± 5.8	94.9 ± 1.6	<0.001 **
HR (initial)	94 ± 24	94.6 ± 24.8	0.824
HR (maximal)	124.9 ± 24.6	128.7 ± 25.4	0.134
CAT score	12.9 ± 7.9	11 ± 6.5	0.013

Abbreviations: FEV1, forced expiratory volume in 1st second; FVC, forced vital capacity; DLCO, diffusion capacity of lung for carbon monoxide; MRC score, medical research council dyspnea scale; MIP, maximum inspiratory pressure; MEP, maximum expiratory pressure; 6MWD, 6 min walking test distance; SpO2, saturation of oxygen; HR, heart rate; CAT, COPD assessment test. ** Adjusted multivariate logistic regression analysis of the ED and Non-ED Group showed that only SpO_2_ (minimal) has difference between the two groups (*p*-value < 0.001).

**Table 3 medicina-59-00391-t003:** Medical Burden and Index of the ED and Non-ED Group.

	ED Group (*n* = 231)	Non-ED Group (*n* = 165)	*p*-Value
ADO index	5.2 ± 1.8	4.6 ± 1.7	<0.001
ADO quartile			0.005
quartile 1	16	24	
quartile 2	68	56	
quartile 3	89	63	
quartile 4	58	22	
ADO quartile 1–2	84	80	0.016
ADO quartile 3–4	147	85	
BODE index	3.3 ± 2.1	2.6 ± 1.9	<0.001
BODE quartile			0.004
quartile 1	98	90	
quartile 2	60	49	
quartile 3	52	19	
quartile 4	21	7	
BODE quartile 1–2	158	139	<0.001
BODE quartile 3–4	73	26	
GOLD stage			
GOLD I	28	18	<0.001
GOLD II	87	100	
GOLD III	96	44	
GOLD IV	20	3	
GOLD I-II	115	118	<0.001
GOLD III-IV	116	47	
Charlson comorbidity index (CCI)	3.5 ± 2.9	3.1 ± 2.6	0.65
Medical burden (NTD)			
number of outpatient visits	20.5 ± 26.5	17.9 ± 11.1	0.92
outpatient medical expenses	67,321.6 ± 98,061.5	59,468.2 ± 78,933.5	0.45
number of hospitalizations	1.1 ± 1.8	0.81 ± 1.6	0.15
number of hospitalization days	12.7 ± 26.9	7.3 ± 14.9	0.13
hospitalization medical expenses	29,288.1 ± 65,578.2	33,698.5 ± 92,445.6	0.36
total medical expenses	96,609.8 ± 121,410	93,166.7 ± 124,076.3	0.41

Abbreviations: ADO index, composite index of age, dyspnea, and obstruction of airflow; BODE index, composite index of body mass index, obstruction of airflow, dyspnea, and exercise capacity. NTD, New Taiwan Dollar.

## Data Availability

The data supporting this research are available from S.-F.L. or C.-L.L.

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
