# Peer review of "The Relationship between Exertional Desaturation and Pulmonary Function, Exercise Capacity, or Medical Costs in Chronic Obstructive Pulmonary Disease Patients"

_medicina, 2023, doi:10.3390/medicina59020391_

Round 1

Reviewer 1 Report

This study aimed to describe the features of patients with COPD with ED, and compare the medical costs between ED COPD group and non-ED COPD group.

1. Is the definition of ED is well established?

2. What is the most related feature to ED?

3. Why was there no difference in medical expenses between the two groups?

4. The medication and adherence may affect the results of medical expenses. 

5. Could the multivariable analysis on ED be of value to provide the more information on the phenotype of ED?

6. When the categorization of three group such as the lowest, middle, highest based on the severity of ED, is no difference between the groups on the medical expenxes found?

Reviewer 2 Report

Thanks for the opportunity to review this work. I have the following comments that hopefully can improve this publication. 

1.- the concept of exertional desaturation is not to my knowledge a common use measurement in general COPD care. most of the data publish in the recent years was generated by some of the authors. I feel is important to explore this phenomenon of exertional desaturation in the introduction to put the current work in better context. 

2.- What are the clinical implications of exertional desaturation.? It will be more interesting to see if these patients that experience exertional desaturation and dont qualify for oxygen behave different than those that qualify for oxygen with activity. 

3,- the adjusted multivariable logistic regression showed that the only difference between the groups was the minimal SpO2. the authors make that a relevant point in the discussion and conclusions but i would expect that to be different as that is the criteria to divide the groups those whose SpO2 is below 90 or drop 4 points. so overall my impression is that separating the patients by ED does not add much to our understanding of outcomes. 

4.- the authors describe multiple possibilities why the above describe difference and all are possible including co-morbidities, etc. but if possible if the have the data it will be intersting to compare the degree of emphysema that the ED group has compare to non ED that could be explanation and will help understanding the DLCO. ( echo data and Ph data will be helpful too)

5.- there is mention in the manuscript of figure 2 but I did not found fig 2 on the document. 

Reviewer 3 Report

Tsai et al. conducted a retrospective study by comparing exertional desaturation (ED) with standard diagnostic/prognostic parameters in COPD. The authors clearly described inclusion and exclusion criteria of the studied patient. ED is not novel in this field. Some comparisons have been performed in larger cohorts, with more information published in those studies. However, one of the major strengths of the study is that the investigations not only focus on the COPD burden for the patients but also on the medical expenses, which has significant implications for the health care system. This aspect was largely ignored and this study may raise attention for further studies investigating the socio-economic impact of COPD as a global disease.

Nevertheless, some aspects can be further explored to provide more meaningful and valuable conclusions.

Comments:

1.    COPD comprises several disease characteristics, such as chronic bronchitis, small airway diseases and pulmonary emphysema. The authors provided information not only on the FEV1/FVC, but also on the diffusion capacity of the lung for carbon monoxide (DLCO). The results showed that DLCO in the ED group significantly decreased compared with the Non-ED group (Table 2). Does it mean that COPD patients with a predominant emphysema endotype constitute mainly the ED group? Do authors have computed tomography (CT) scans to stratify different phenotypes (e.g. COPD with predominant chronic bronchitis or with predominant emphysema) between/within ED and non-ED groups?

2.    In Table 3, it is worth noting that there is a marked difference between ED and non-ED groups at BODE quartiles 3 and 4. Do the authors have the results from these human subjects' echocardiography or right heart catheterization (RHC)? As COPD patients often suffer from pulmonary hypertension, could authors address (if the results from echocardiography/RHC are available) or speculate whether ED could be an indicator for COPD patients with additional pulmonary hypertension?

3.    Is there a significant difference in the GOLD stage between ED and non-ED patients?

Minor comments:

1.    On page 2, line 91, a word might be missing. … diagnosis of COPD by a post-bronchodilator “test”.

2.    Is the unit of outpatient medical expenses in Table 3 also New Taiwan Dollar? If so, please add it. And please address the abbreviation of NTD.

3.    Could it be a possible error occurring on page 6 lines 176 and 177?

4.     On page 7, line 225. It is, in general, ambiguous to say Western countries. Could authors replace the term with, for example, “various nations and states in the regions of Europe, North America, and Oceania”?

5.    Page 7, line 242. Is the dot meaningful in front of SpO2?

Round 2

Reviewer 1 Report

All the comments have been mostly addressed.

Reviewer 2 Report

authors have clarify all my points.

thanks